# The Impact of the COVID-19 Pandemic on Pediatric Bicycle Injury

**DOI:** 10.3390/ijerph20085515

**Published:** 2023-04-14

**Authors:** Gates R. L. Failing, Brett G. Klamer, Tyler J. Gorham, Jonathan I. Groner

**Affiliations:** 1University Hospitals Cleveland Medical Center, Cleveland, OH 44106, USA; 2Center for Biostatistics, Department of Biomedical Informatics, The Ohio State University, Columbus, OH 43210, USA; 3Biostatistics Resource at Nationwide Children’s Hospital, Columbus, OH 43205, USA; 4IT Research & Innovation, Nationwide Children’s Hospital, Columbus, OH 43205, USA; 5Department of Surgery, Nationwide Children’s Hospital, Columbus, OH 43205, USA; jonathan.groner@nationwidechildrens.org; 6College of Medicine, The Ohio State University, Columbus, OH 43210, USA

**Keywords:** bicycle, injury, COVID-19, pediatric trauma, time series, pediatrics, safety, injury prevention

## Abstract

Bicycling is a common childhood activity that is associated with significant injury risk. This study’s aim was to assess pediatric bicycle injury epidemiology and impacts of the COVID-19 pandemic. We conducted a cross-sectional evaluation of patients age < 18 years presenting with bicycle injury to a pediatric trauma center. A pre-pandemic period (1 March 2015–29 February 2020) was compared to the pandemic period (1 March 2020–28 February 2021). A total of 611 injury events for children < 18 years were included (471 pre-pandemic events and 140 pandemic events). The relative frequency of pandemic injuries was greater than pre-pandemic injuries (*p* < 0.001), resulting in a 48% increase in pandemic period injuries versus the pre-pandemic average (141 pandemic vs. 94.4/year pre-pandemic). Individuals of female sex represented a larger proportion of injuries in the pandemic period compared to the pre-pandemic period (37% pandemic vs. 28% pre-pandemic, *p* = 0.035). Injuries were more common on weekends versus weekdays (*p* = 0.01). Time series analysis showed a summer seasonality trend. Localizing injury events to ZIP codes showed regional injury density patterns. During COVID-19, there was an increase in bicycle injury frequency and proportional shift toward more injuries involving individuals of female sex. Otherwise, injury patterns were largely unchanged. These results demonstrate the necessity of safety interventions tailored to community needs.

## 1. Introduction

Worldwide, bicycling is a ubiquitous childhood pastime that fulfills recreation, fitness, and transportation needs. Among children ages 6 to 17 in 2018, an estimated 11 million rode a bicycle at least once in the USA [1]. However, bicycles can have serious consequences for children. There were an estimated 108,124 nonfatal emergency department visits and 59 fatalities in the USA for bicyclists ages 18 or younger in 2018 [2]. Pediatric bicycle injury has previously been studied to assess risk factors and injury severity. Generally, males ages 10 to 14 years have the highest rate of bicycle injury based on representative data from the National Electronic Injury Surveillance System during the 1990–2005 time period [3] as well as the 2006–2015 time period [4]. Extremity lacerations and contusions were the most common types of injuries. Given the associated morbidity and mortality, head injury is a focal point for research. Between 2006 and 2015, head and neck injury accounted for almost 15% of all bodily injury locations, and traumatic brain injury (TBI) accounted for almost 11% of all injury types, most commonly among ages 10 to 14 [4]. 

Bicycle helmets are a major intervention for injury prevention. According to a metanalysis of five studies, helmets reduce head injury by an estimated 48% among all age levels [5]. Similarly, a retrospective analysis of the National Electronic Injury Surveillance System from 2006 to 2015 corroborated these findings, demonstrating that bicycle helmet use significantly reduced the likelihood of head and neck injury and hospitalizations [4]. However, there is debate in the adult bicycle injury literature that helmets may paradoxically encourage riskier behavior, termed the risk compensation theory [6,7], both by motor vehicle drivers while passing bicyclists [8] and by helmeted individuals under simulated conditions [9]. More recent research disputes this theory, namely a reanalysis of the Walker 2007 study which found that reductions in passing distance by motorists were not strongly associated with use of helmets among bicyclists [10]. Additionally, intersection camera analysis found that other than the speed variable, helmeted bicyclists were no more risky than unhelmeted bicyclists [6]. A systematic review of 23 articles found only two studies that supported the risk compensation theory [7], emphasizing the protective nature of bicycle helmets rather than any propensity for behavioral changes. While most helmet behavioral studies have been conducted in adults rather than children, the American Academy of Pediatrics unequivocally endorses helmet use for injury prevention [11]. 

Demographic and urban planning factors have been correlated with bicycle injury. A study using census tracts in Austin, TX found that poverty rates were associated with more frequent trips by bicycle and, consequently, more frequent accidents. In the same study, increases in community sidewalks and public transit stops were associated with increased injury rates, which were attributed to increased rates of utilization [12]. A single-center study of 77 pediatric cases using geographic and demographic data found clustering of bicycle injury events around major thoroughfares and lower income communities [13]. Other studies have highlighted the importance of multiple methodologies to model bicycle injury. Time series analysis is a tool for evaluating bicycling patterns in response to an external event, such as before and after a hurricane [14]. Large cities have used injury density mapping to develop bicycle safety interventions in specific regions/ZIP codes. A study in New York City mapped bicycle infrastructure and injury density over time to determine the efficacy of injury prevention interventions [15].

Time period and social changes can also affect bicycle injury rates. The COVID-19 pandemic upended traditional recreation patterns and, by extension, pediatric injury presentations. A retrospective cohort study of 1745 patients at a level 1 pediatric trauma center during COVID-19 found a significant decrease in acute fracture presentations (attributed to changes in play patterns), but also reported an increase in the rate of injury for self-powered vehicles, like bicycles, in comparison to the same time period during non-pandemic years [16]. Analysis of a national database in Canada for pediatric emergency visits reported an increase in bicycle injuries during COVID-19 that was on par with 2015 levels [17]. Very few single-center studies focused solely on pediatric bicycle injury during the pandemic. A cross-sectional study of 1215 injuries at a tertiary children’s hospital in Canada reported a significant increase in monthly bicycle injuries during the COVID-19 period compared to the same period in 2018–2019 [18]. Similarly, a tertiary center in Australia also reported a significant increase in pandemic bicycle injury presentations compared to the year prior [19]. However, a California trauma center found that bicycle injury admissions decreased by 28.4% during the pandemic compared to the previous year [20]. Thus, there is likely significant center-to-center and year-to-year variability when studying pediatric bicycle injury, particularly when the comparison period is only one or two years in length. 

In this study, we sought to assess the COVID-19 pandemic’s impact on the epidemiology of pediatric bicycle injury, including demographics, frequency, location, and severity. Secondly, we sought to assess cyclical and seasonal variations in bicycle injury over time. Finally, we mapped bicycle injuries to specific ZIP codes in order to better understand the distribution of bicycle injury event densities.

## 2. Materials and Methods

This is a retrospective review of a trauma registry from a large urban level 1 pediatric trauma center. It is the sole pediatric trauma center within the region and receives the vast majority of children with significant injuries. Inclusion criteria included patients younger than 18 years of age with a bicycle-related injury based on the International Classification of Disease, tenth edition (ICD-10) codes V10–V19, either as a direct presentation or outside hospital transfer. Data were abstracted from the trauma registry and then sorted and coded for analysis. Sex and race were self-reported by patients and their families. The Abbreviated Injury Scale was used to quantify head injury, ranging from 1 (minor) to 6 (maximum). The Injury Severity Scale (ISS) was used to quantify full body injury by summing the squares of the three highest AIS scores based on the following body regions: head and neck, face, chest, abdominal, and external [21].

The pandemic period (1 March 2020–28 February 2021) was compared to the same time interval over the previous five years to form a pre-pandemic period (1 March 2015–29 February 2020). The date for the start of the pandemic was based on COVID-19 confirmed cases in the United States and public health declarations by the World Health Organization and US government in early March 2020 [22]. The date for the end of the pandemic period was based on local trends in COVID-19 cases and stay at home orders in the catchment area of the study location. Descriptive data were summarized as frequencies and percentages for categorical variables and medians and interquartile ranges (IQR) for continuous variables. Fisher’s exact tests were used to determine statistically significant differences by pandemic groups (pre-pandemic versus pandemic) for categorical variables. Wilcoxon rank sum tests were used to determine statistically significant differences by pandemic groups for continuous variables. A chi-squared goodness-of-fit test was used to test for difference in injury frequency by day of week (weekday versus weekend) and rolling 12-month time period.

Bicycle injury events from 1 January 2015 to 28 February 2021 were aggregated at the monthly level and used for time series analysis. We modeled this count data using an integer autoregressive conditional heteroscedastic (INGARCH) model assuming the conditional distribution of the time series at time t given the history of the joint process at time t−1 as negative binomial with logarithmic link function [23]. We accounted for seasonality by regressing on the conditional mean 12 months back in time and serial dependence by regressing on the previous month’s observation. An indicator variable for the pandemic time period which assumed a constant effect forward in time and a linear trend term was also included. Both covariates were fit as internal effects, meaning that the influence of each propagated to future observations by regression on both the past observations and the past conditional means. A score test was used to test for significant intervention effect of the pandemic (1 March 2020–28 February 2021) on expected counts [24]. Predicted injury counts with corresponding 90% prediction intervals are provided for the 1 March 2021–28 February 2022 time period assuming the pandemic effect had subsided. *p* values less than 0.05 were considered statistically significant. Analysis was conducted using version 4.1.3 of R software [25] including the tidyverse [26], tidycensus [27], and tscount [23] packages.

For spatial analyses, bicycle injuries were aggregated to the ZIP code of injury location. ZIP codes were chosen as the unit of analysis over census tracts due to the relatively small number of injuries in the study population. Because annual population estimates are not available at the ZIP code level, we relied on the 2016–2020 American Community Survey (ACS) five-year estimates [28] for population and demographic estimates, as this five-year interval coincides with the center of our study period. Injury rates were smoothed using spatial Empirical Bayes smoothing with a first-order queen contiguity matrix in version 1.20.0.8 of GeoDa software [29] and plotted using the tmap package in R software [30]. The most likely injury rate cluster was identified using Martin Kulldorff’s spatial scan statistic, executed using R’s SpatialEpi package, version 1.2.7, with a maximum population size of 10%, 999 Monte Carlo simulations, and an alpha level of 0.05 [31,32].

## 3. Results

Between 1 January 2015 and 28 February 2021, 612 bicycle injury events were identified from the trauma registry that met our inclusion criteria. Of these, 472 occurred before 1 March 2020, and 140 after. To compare patient characteristics before and during the pandemic, we defined the pre-pandemic baseline period as 1 March 2015–29 February 2020 which included 471 events, and the pandemic period as 1 March 2020–28 February 2021 which included 140 injury events for a total of 611 injury events as shown in Table 1. 

There was a 48% increase in the frequency of bicycle injuries for the pandemic period (*N* = 140) versus the rolling 12-month pre-pandemic periods (mean = 94.4; range: 78, 106; *p* < 0.001; Table 1). The prominent seasonality effect where injury events peak in July and reach a minimum in December and January is shown in Figure 1. Weekends (Saturday and Sunday) had a higher frequency of injuries compared to weekdays (*p* = 0.01; Figure 2). 

The pandemic period was associated with a decrease in percentage of injuries involving individuals of male sex (72% pre-pandemic vs. 63% pandemic) and an increase in percentage of injuries involving individuals of female sex (28% pre-pandemic vs. 37% pandemic) when comparing the two periods (*p* = 0.035; Table 1). Notably, there were no other statistically significant differences in age, race, injury location, trauma level, protective device, head injury frequency, AIS head scores, or ISS scores for the two periods (Table 1).

The count-based time series model showed increased expected injury counts during the pandemic time period (*η* = 0.29, *p* = 0.54). The multiplicative effect size of the pandemic is exp(0.29) = 1.34, resulting in an estimated 34% increase in injuries during the pandemic. Although the effect of the pandemic was not found to be significant using this time series model, this does not rule out a pandemic effect, since results were found to be sensitive to model assumptions. Figure 3 shows the time series of observed injury events and model predictions (with 90% prediction interval) assuming the pandemic effect has been reduced to zero for March 2021 to February 2022.

Mapping injuries across Franklin County, Ohio between 1 January 2015–28 February 2021 showed regions of elevated bicycle injury density with the greatest injuries per 10,000 children per year on the southwest side of the county (Figure 4). 

## 4. Discussion

During the pandemic, there was a 48% increase in bicycle injuries compared to the average over the five non-pandemic years. We chose to use a five-year non-pandemic comparison period to reduce the impact of year-to-year fluctuations in injury rates when compared to the COVID-19 time period. Our observed percent increase corresponds with the predicted effect of a 34% increase in injuries during the pandemic period based on the time series model. The observed increase likely reflects changes in recreation patterns during the COVID-19 pandemic which are supported by other studies showing increases in bicycle injury during this period [16,17,18,19]. One study of COVID-19 pediatric trauma admissions at a level 1 center used a novel approach to differentiate between the pandemic period and the subsequent post-pandemic reopening period based on local guidelines. They found relative percentage increases in self-powered transportation-related injuries in the latter two periods [33], which is in alignment with the results of our present study. However, evidence from parent reports during COVID-19 suggests that physical activity among children declined and sedentary behaviors increased, including those associated with screen use [34]. This was likely due to school closure and cancellation of organized recreational activities. Nevertheless, the physical activity children did pursue due to increases in leisure time may have been centered on informal home or neighborhood-based activities where bicycling may have been more common [19].

In our study, the percentage of injuries involving individuals of female sex increased and the percentage of injuries involving individuals of male sex decreased during the pandemic compared to the pre-pandemic time period. Corroborating these results, a study of pediatric fractures before and during COVID-19 found a proportional decrease in fractures involving children of male sex and a proportional increase in fractures involving females, though it was not statistically significant [16]. However, most nationally representative pediatric injury research has shown that children of male sex are at higher risk for bicycle injury [3,4], and other recent single-center studies showed that males continued to make up the majority of bicycle injury events during the pandemic [18,19]. Child psychology research suggests that children of female sex self-appraise bicycle injury risk as greater compared to males when looking at photographs of risky bicycle behavior scenarios [35]. One potential explanation for our observed trend is that COVID-19 school cancellations and social distancing may have altered the traditional risk appraisal calculus for children and encouraged riskier recreational activities. In support of this theory, one study found that all-terrain vehicle (ATV) injuries, a high-risk recreational activity, increased by 78% in the pandemic period compared to pre-pandemic periods [33]. These results highlight the importance of focusing intervention efforts on cyclists who identify as female to provide targeted education and rider empowerment.

No significant changes were observed in helmet use when comparing the pandemic versus the pre-pandemic periods. One explanation could be the presence of a substantial proportion of missing data for helmet use (Table 1). Nevertheless, the percentage of non-classified data for protective devices was lower during the pandemic period (10/140) compared to the pre-pandemic period (80/471), suggesting that the act of reporting helmet use may have improved during the pandemic. In prior studies, helmet use documentation was only present in about 11% of total events [4] and 9% of total events [36], which limits the statistical power of comparisons. Otherwise, there were few differences among injury variables in the dataset when comparing the pre-pandemic and pandemic periods (Table 1), suggesting that bicycle injury patterns were largely constant at our trauma center despite massive societal changes during the COVID-19 pandemic.

The time series analysis suggested the presence of seasonal variation on a monthly and weekly basis. Warmer months during summer in our study location within the Northern Hemisphere had higher percentages of bicycle injuries. These results are in alignment with another time series analysis of urban bicycle and motor vehicle accidents that showed peaks during summer months and declines during winter months [37]. During the summer, school closure and warmer weather encourage outdoor recreational activities, hence increasing the volume of bicyclists. However, seasonal variation is dependent on location. A study in Florida found that most injuries occurred in spring and winter, which are correlated with more moderate temperatures and likely increased rates of outdoor activities like bicycling [36]. Weekend days, specifically Saturday and Sunday, also had the highest percentages of bicycle injuries in our study which are periods that typically have ample recreation time in the USA.

Mapping injury locations demonstrated the highest injury density in the southwest region of Franklin County. Several hypotheses could explain these results. There may be differences in bicycle infrastructure in these ZIP codes, as prior research suggests that fewer bicycle lanes, more intersections, fewer streetlights, and higher speed limits are associated with more bicycle injuries [38]. Perhaps certain ZIP codes in Franklin County have road or traffic elements that are especially hazardous to bicyclists. Additionally, there may be differences in socioeconomic status between areas with higher and lower bicycle injury densities. Yu (2014) suggested that more affluent areas in Austin, TX have more pedestrian resources and are less dependent on bicycles for transportation whereas indigent areas are more dependent on bicycles for transportation to work but lack the infrastructure necessary for sufficient safety [12]. Variation in utilization could also be responsible for the observed pattern. For example, residents of certain ZIP codes may have greater access to organized after school activities (e.g., music, sports), so informal activities like bicycling are less frequently pursued compared to other areas. Additionally, bicycle ownership among youth may also vary among ZIP codes. Future studies are necessary to analyze factors in these ZIP codes that are drivers of bicycle injury.

There were several limitations to our study. Firstly, the data were only from a single institution in an urban Midwest region of the United States. Bicycle injury patterns might be different at facilities with a more rural catchment area or in areas with a warmer and/or colder climate. Previous research from 2000 to 2014 demonstrated significant state-to-state variability in bicycle accident rates, with the highest being in California and the lowest in Arkansas [39]. Future studies should assess the effect of the COVID-19 pandemic on bicycle injury in rural locations and in additional US states to see if there is concordance with our results. Using demographic data from the American Community Survey is another limitation, as population data points are based on probability sampling, which is subject to sampling error [40]. Secondly, there was a relatively high proportion of non-classified data that was recorded as “N missing” in Table 1 for several variables, especially protective devices and injury location, which could have skewed elements of the analysis. Additionally, there may have been a general underreporting of bicycle injury during the pandemic, as individuals were more reticent to enter healthcare facilities due to the perceived risk of virus transmission. Several assumptions were made for the time series analysis, including that the bicycle injury counts were not biased significantly by the different number of days by month, different number of weeks by month, unequal number of weeks per year, and the number of holidays and long weekends in a year. Additionally, the pandemic was assumed to start on 1 March 2020, and to have a constant effect on injury rates until 28 February 2021 [22]. The actual pandemic effect is likely to have been a transient shift such as that described by exponential decay or more complex nonlinear processes.

## 5. Conclusions

The impact of the COVID-19 pandemic was an overall increase in bicycle injuries and an increased proportion of total bicycle injuries among individuals of female sex. However, our study did not show other major changes to pediatric bicycle injury statistics due to the COVID-19 pandemic. Conclusions from our study were limited by study population location, biases within demographic data, a high proportion of non-classified data for certain variables, and assumptions regarding pandemic variables. Additional studies should examine other centers and regions of the US to determine if the pandemic significantly affected pediatric bicycle injury. Data from this study can inform future global pandemics to emphasize targeted injury prevention efforts, from anticipatory guidance in the pediatrician’s office to community wide helmet distribution campaigns. Additionally, there is an urgent need for interventions that target vulnerable populations given the morbidity associated with injury as well as the importance of bicycles in indigent communities. Underserved neighborhoods disproportionately bear the burden of bicycle injury [12]. Two of the authors of our paper previously conducted a community educational intervention in Franklin County to target road riding safety among adults and children. In partnership with a local bicycle nonprofit, we designed a series of community rides to build camaraderie and teach safe cycling behaviors like signaling, lane positioning, and navigating in an urban area with limited bicycle lanes. Community members were receptive to the program, which achieved success prior to the arrival of COVID-19. Future efforts must be initiated in other high-risk ZIP codes, particularly those identified in this study, to provide practical, accessible, and affordable cycling and pedestrian education to children and families.

## Figures and Tables

**Figure 1 ijerph-20-05515-f001:**
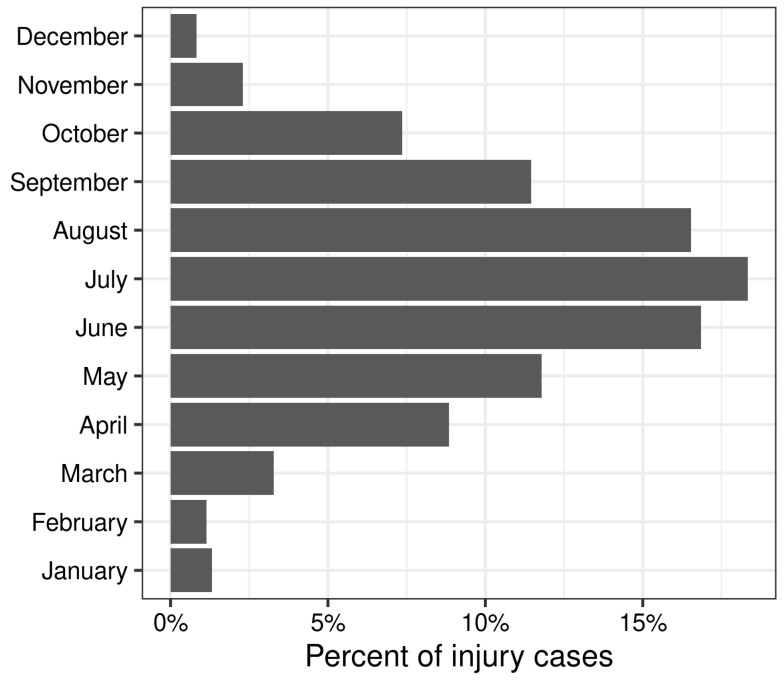
Percentage of total injury counts per month. Aggregated over the 1 March 2015–28 February 2021 time period.

**Figure 2 ijerph-20-05515-f002:**
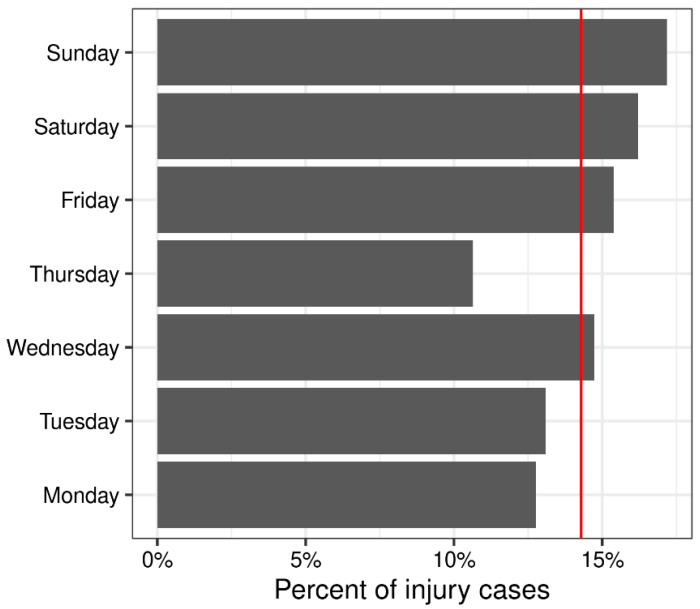
Percentage of total injury counts per day of the week. The red line shows the expected percentage assuming no difference by day of the week. Aggregated over the 1 March 2015–28 February 2021 time period. Injuries were more likely on weekends compared to weekdays (*p* = 0.01).

**Figure 3 ijerph-20-05515-f003:**
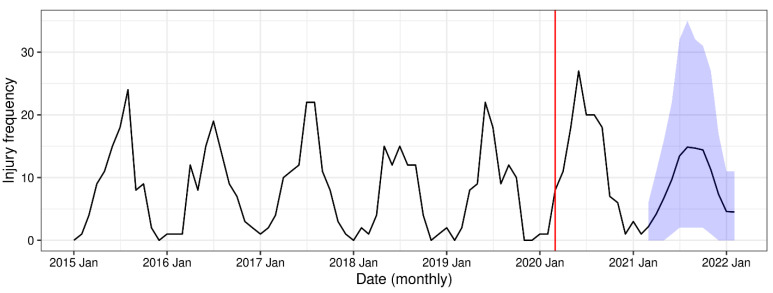
Monthly frequency of pediatric bicycle injuries between 1 January 2015 and 30 March 2022. Start of COVID-19 pandemic on 1 March 2020 is shown with the red line. Blue shaded area is 90% prediction interval for pediatric bicycle injury, assuming the pandemic effect had decayed to zero, from 1 March 2021 to 28 February 2022.

**Figure 4 ijerph-20-05515-f004:**
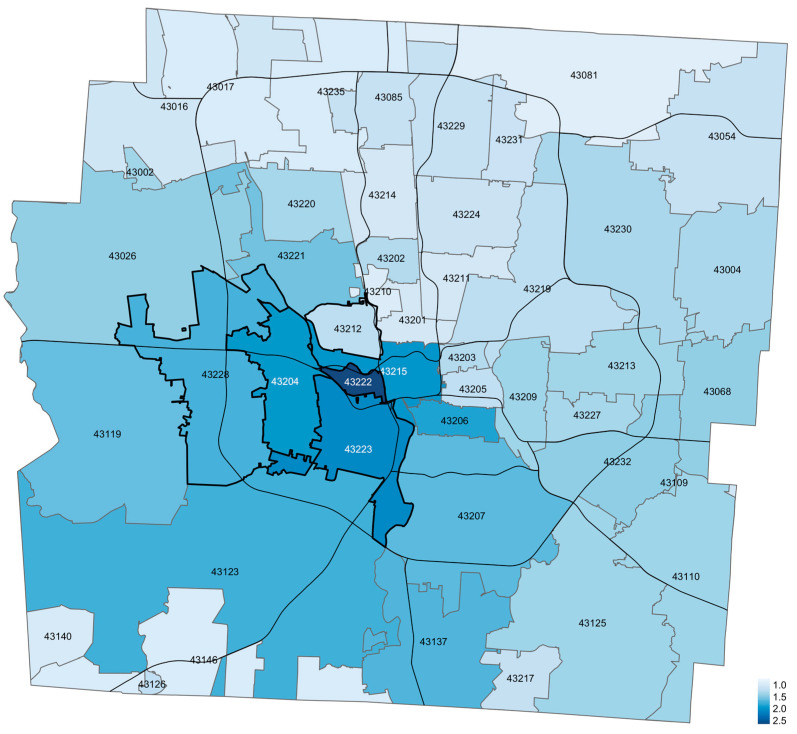
Franklin County, Ohio Bicycle Injury Density Map from 1 January 2015 to 28 February 2021. Units are injury events per 10,000 children per year, all years (Smoothed). ZIP codes are noted with numbers and borders.

**Table 1 ijerph-20-05515-t001:** Comparison of patient characteristics between pre-pandemic and pandemic time periods.

Characteristic	Pre-Pandemic, *N* = 471 ^1^	Pandemic, *N* = 140 ^1^	*p*-Value ^2^
Age (years)	10.0 (7.0, 12.0)	10.0 (7.0, 12.0)	0.8
Sex			0.035
Male	341 (72%)	88 (63%)	
Female	130 (28%)	52 (37%)	
Race			0.4
White	354 (77%)	108 (77%)	
Black	72 (16%)	17 (12%)	
Asian	13 (2.8%)	3 (2.1%)	
Hispanic	13 (2.8%)	6 (4.3%)	
Other	10 (2.2%)	6 (4.3%)	
(N Missing)	9	0	
Injury location			0.7
Home Residence	71 (19%)	21 (21%)	
Street/Parking lot/Driveway	253 (69%)	70 (69%)	
Recreational/Sports area	26 (7.1%)	8 (7.9%)	
Other	17 (4.6%)	2 (2.0%)	
(N Missing)	104	39	
Trauma level			0.10
Level 1 Trauma	43 (9.1%)	14 (10%)	
Level 2 Trauma	187 (40%)	43 (31%)	
Level 1 Neuro trauma	7 (1.5%)	0 (0%)	
No trauma alert activation	234 (50%)	83 (59%)	
Protective devices			0.9
Helmet worn	61 (16%)	19 (15%)	
Helmet not worn	330 (84%)	111 (85%)	
(N Missing)	80	10	
Head injury			0.2
No head injury present	342 (73%)	94 (67%)	
Head injury present	129 (27%)	46 (33%)	
AIS			0.2
1	41 (33%)	10 (23%)	
2	36 (29%)	18 (41%)	
3	32 (25%)	11 (25%)	
4	9 (7.1%)	5 (11%)	
5	8 (6.3%)	0 (0%)	
(N Missing)	345	96	
ISS	4.0 (1.0, 6.0)	4.0 (3.5, 8.2)	0.066
(N Missing)	7	0	

^1^ Median (IQR); *n* (%), ^2^ Wilcoxon rank sum test, Fisher’s exact test, AIS = abbreviated injury scale, ISS = injury severity scale; Pre-pandemic = 1 March 2015 to 29 February 2020, Pandemic = 1 March 2020 to 28 February 2021.

## Data Availability

Data is not publicly available due to the sensitive nature of protected patient information.

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
