# Peer review of "The Impact of the COVID-19 Pandemic on Pediatric Bicycle Injury"

_ijerph, 2023, doi:10.3390/ijerph20085515_

Round 1

Reviewer 1 Report

The authors of the manuscript (IJERPH-2259663), entitled “The impact of the COVID-19 pandemic on pediatric bicycle injury,” tend to apply cross-sectional evaluation to assess pediatric bicycle injury epidemiology and impacts of the COVID-19 pandemic. According to the title, it is expected that the authors can identify some key criteria which can be used to compare the injury in the pre-pandemic period and pandemic period in a fair and objective ground. However, this research failed to do so. Without a solid research ground, various of analysis methods become meaningless.

Author Response

Reviewer #1

Point 1:

The authors of the manuscript (IJERPH-2259663), entitled “The impact of the COVID-19 pandemic on pediatric bicycle injury,” tend to apply cross-sectional evaluation to assess pediatric bicycle injury epidemiology and impacts of the COVID-19 pandemic. According to the title, it is expected that the authors can identify some key criteria which can be used to compare the injury in the pre-pandemic period and pandemic period in a fair and objective ground. However, this research failed to do so. Without a solid research ground, various of analysis methods become meaningless.

-Response 1:

The authors respectfully disagree with the statement that the title (“The impact of the COVID-19 pandemic on pediatric bicycle injury”) implies that “the authors can identify some key criteria which can be used to compare the injury in the pre-pandemic period and pandemic period in a fair and objective ground.”  The authors chose this title because of the anecdotal observation that more bicycle injuries were occurring among children during the pandemic shutdown.  We initiated this research project because we were seeking to find objective data to see if the COVID-19 pandemic did, in fact, impact the number of children per unit time (the rate) of bicycle injuries increased during the lockdown.  Our statistical analysis showed that there was an overall increase in bicycle injuries and an increased proportion of total bicycle injuries among individuals of female sex during the pandemic period, but we did not show other major changes to pediatric bicycle injury statistics due to the COVID-19 pandemic. This title change was the only specific item mentioned by the reviewer in the comments section, and no other written comments were provided.

Reviewer 2 Report

The authors have explored a very interesting topic. However, I suggest a minor revision before publishing the paper.

In the abstract, is p=0.01 correct, or should it be p<0.01

Please see the guidelines for authors (citing references in the text, figure title, etc.).

The authors have an interesting introduction. However, I suggest that you include more recent references in the introduction and touch on the aspect that Volker and his collaborators analyzed and similar works:

10.1371/journal.pone.0075424

https://doi.org/10.1016/j.trf.2020.09.014

https://doi.org/10.3390/app13031621

The methodology is clearly written.

How did you resolve the cut-off dates for the pandemic to appear in certain areas?

For these values in parentheses (e.g. (63% vs. 72%)) does the first value apply before the pandemic and the second, after the pandemic? Please unify and reformulate this.

I ask you to arrange the paper technically. Also, although I am not competent in English, I came across a couple of sentences that should be reworded.

I ask you to expand the conclusion section with the limitations of the study, future directions of research, and application of the obtained results in practice.

Author Response

Reviewer #2

The authors have explored a very interesting topic. However, I suggest a minor revision before publishing the paper.

Point 1: In the abstract, is p=0.01 correct, or should it be p<0.01.

-Response 1: P=0.01 is correct.

Point 2: Please see the guidelines for authors (citing references in the text, figure title, etc.).

Response 2: Citations were updated.

Point 3: The authors have an interesting introduction. However, I suggest that you include more recent references in the introduction and touch on the aspect that Volker and his collaborators analyzed and similar works:

10.1371/journal.pone.0075424

https://doi.org/10.1016/j.trf.2020.09.014

https://doi.org/10.3390/app13031621

-Response 3: A section was added in the introduction addressing this and relevant citations were added.

The methodology is clearly written.

Point 4: How did you resolve the cut-off dates for the pandemic to appear in certain areas? 

-Response 4: A statement was added in the methods section addressing this.

Point 5: For these values in parentheses (e.g. (63% vs. 72%)) does the first value apply before the pandemic and the second, after the pandemic? Please unify and reformulate this.

-Response 5: Adjustments were made to this in results section to make it clear which percent referred to which time period.

Point 6: I ask you to arrange the paper technically. Also, although I am not competent in English, I came across a couple of sentences that should be reworded.

-Response 6: The authors stand by their statistical methods which are commonly used in this type of research involving trauma registry data and time series analysis. Edits were made in methods section.

Point 7: I ask you to expand the conclusion section with the limitations of the study, future directions of research, and application of the obtained results in practice.

-Response 7: The conclusion section was expanded.

Round 2

Reviewer 1 Report

The authors try to use simple comparison to find the“The impact of the COVID-19 pandemic on pediatric bicycle injury.”However, there are many possible reasons (factors) which could affectthe outcome. It is very inappropriate to use rough statistical analysis to claim that everything happened during the pandemic period of time is 100% affected by COVID-19 in an academic research. The value of a sound research should be based on a complete research design which can control variables as many as possible so that the real impact can be demonstrated. 

Author Response

Our study, like the vast majority of the published literature involving real-world injury data, cannot be performed “based on a complete research design which can control variables as many as possible.”  Randomized controlled trials in trauma research are not only impossible but unethical.  For example, it would be unethical to give a randomized group of children bicycle helmets and compare their injury rate to a control group without helmets.  Likewise, it is not possible (and indeed not ethical) to expose one population to the effects of the COVID-19 pandemic and compare cycling injures with another population that was not exposed.

Therefore, our study represents the next best thing:  we examined a population in the same geographic area and compared the injury rate before and after a major environmental change:  the COVID-19 pandemic.  Although it is true that other environmental factors could have impacted the number of bicycle injuries (changes in traffic patterns or traffic density, for example), this seems unlikely.  The COVID-19 pandemic impacted every aspect of daily living here in Columbus:  it shut down schools, closed businesses, decreased car traffic, canceled major outdoor events, and severely limited vacation travel.  Thus, it is highly probable that the changes in bicycle injury patterns described in the article can be attributed to the pandemic.

Thus, the authors stand by the methodology used in their paper and would be happy to provide published research articles by other authors using similar methods. Many of the peer-reviewed articles assessing pediatric injury before and during the COVID-19 pandemic, as cited in the introduction of the present manuscript, use similar methodologies:

Bram, J.T.; Johnson, M.A.; Magee, L.C.; Mehta, N.N.; Fazal, F.Z.; Baldwin, K.D.; Riley, J.; Shah, A.S. Where have all the fractures gone? The epidemiology of pediatric fractures during the COVID-19 pandemic. J Pediatr Orthop 2020, 40 (8), 373–379, doi:10.1097/BPO.0000000000001600.

Keays, G.; Friedman, D.; Gagnon, I. Pediatric injuries in the time of COVID-19. Health Promot Chronic Dis Prev Can 2020, 40 (11-12), 336–341, doi:10.24095/hpcdp.40.11/12.02.

Shack, M.; Davis, A.L.; Zhang, E.W.; Rosenfield, D. Bicycle injuries presenting to the emergency department during COVID-19 lockdown. J Paediatr Child Health 2022, 58 (4), 600–603, doi:10.1111/jpc.15775.

van Oudtshoorn, S.; Chiu. K.Y.C.; Khosa, J. Beware of the bicycle! An increase in paediatric bicycle related injuries during the COVID-19 period in Western Australia. ANZ J Surg 2021, 91 (6), 1154–1158, doi:10.1111/ans.16918.